# Glucose Inhibits Yeast AMPK (Snf1) by Three Independent Mechanisms

**DOI:** 10.3390/biology12071007

**Published:** 2023-07-14

**Authors:** Kobi Simpson-Lavy, Martin Kupiec

**Affiliations:** The Shmunis School of Biomedicine & Cancer Research, Tel Aviv University, Ramat Aviv, Tel Aviv 69978, Israel; kobisimpsonlavy@gmail.com

**Keywords:** Snf1, AMPK, fermentation, respiration, yeast, *Saccharomyces cerevisiae*, hexose, glucose metabolism

## Abstract

**Simple Summary:**

Glucose is the preferred carbon source for most cells. Yeast Snf1 (AMPK in mammals) is the main regulator of the response to low glucose availability. In the last years, much progress has been made in understanding the regulation of Snf1. We summarize three different mechanisms that control Snf1 activity: (1) phosphorylation and dephosphorylation of a critical threonine residue; (2) post-translational modification by the addition of a SUMO tag; (3) protonation and de-protonation of a polyhistidine tract. These mechanisms act independently of each other, allowing a flexible response to the availability of carbon sources.

**Abstract:**

Snf1, the fungal homologue of mammalian AMP-dependent kinase (AMPK), is a key protein kinase coordinating the response of cells to a shortage of glucose. In fungi, the response is to activate respiratory gene expression and metabolism. The major regulation of Snf1 activity has been extensively investigated: In the absence of glucose, it becomes activated by phosphorylation of its threonine at position 210. This modification can be erased by phosphatases when glucose is restored. In the past decade, two additional independent mechanisms of Snf1 regulation have been elucidated. In response to glucose (or, surprisingly, also to DNA damage), Snf1 is SUMOylated by Mms21 at lysine 549. This inactivates Snf1 and leads to Snf1 degradation. More recently, glucose-induced proton export has been found to result in Snf1 inhibition via a polyhistidine tract (13 consecutive histidine residues) at the N-terminus of the Snf1 protein. Interestingly, the polyhistidine tract plays also a central role in the response to iron scarcity. This review will present some of the glucose-sensing mechanisms of *S. cerevisiae*, how they interact, and how their interplay results in Snf1 inhibition by three different, and independent, mechanisms.

## 1. Introduction

Optimization of carbon catabolism for ATP and metabolite production in response to different environmental conditions is key for all organisms. Most cells will prefer glucose as carbon source. This is also true for the yeast *Saccharomyces cerevisiae*. Yeasts must be very careful in regulating glucose metabolism because of the way they metabolize it: when glucose is abundant, they inefficiently ferment it, but when it is scarce, they efficiently oxidize it. The response to the deprivation of intracellular glucose sources is performed in eukaryotes by a kinase complex known as SNF1 (Sucrose non-fermenting 1) in fungi, SnRK (Snf1-related kinase) in plants and AMPK (AMP-dependent kinase) in metazoans, with homologues of the complex’s component proteins present also in proteobacteria and archaea [1]. In fungi, Snf1 retools the metabolic and signaling pathways of the cell in response to an absence of glucose (or fructose) to permit utilization of alternative carbon sources such as galactose, maltose, sucrose, glycerol, lactate, ethanol, acetate, or fatty acids. Snf1 has many targets, including:(1)Transcriptional activators (e.g., Sip4 [2]);(2)Transcriptional inhibitors (e.g., Mig1 [3]);(3)Metabolic enzymes (e.g., Acc1—to inhibit fatty acid synthesis [4], Pfk27, whose degradation following phosphorylation promotes gluconeogenesis [5]);(4)Arrestins (e.g., Rod1), which regulate the degradation of cell membrane proteins such as carbon source importers (e.g., Jen1) [6,7], and other signaling pathways (e.g., inhibition of adenyl cyclase [8]).

Although the complex regulation of Snf1 has become appreciated in the last years, much remains to be explained regarding Snf1 spatial localization, and how it is able to integrate signals from very different inputs.

## 2. Glucose Sensing Pathways in *S. cerevisiae*

Glucose is the preferred carbon source of most organisms. In *S. cerevisiae*, several mechanisms exist to detect glucose. Extracellular glucose is sensed by the plasma membrane localized receptors Snf3 and Rgt2. Snf3 is required for sensing low (0.1%) levels of glucose, whereas Rgt2 is required for the maximal response to high (4%) levels of glucose, as determined by *HXT1* expression [9]. Activated Rgt2 or Snf3 recruits the Yeast Casein Kinases Yck1 and Yck2 to phosphorylate Mth1 and Std1 leading to their degradation. This inactivates the transcriptional repressor Rgt1 by enabling its phosphorylation by PKA and allows expression of glucose transports (especially *HXT1* and *HXT3*) [10,11,12,13,14]. The degradation of Mth1 by glucose is inhibited by high Snf1 activity [15,16], though the mechanism has not yet been determined.

Extracellular glucose (and sucrose, preferentially) binds to the plasma membrane localized Gpr1 receptor resulting in GTP loading of Gpa2. Gpa2^−GTP^ activates Cyr1 (adenyl cyclase) activity to produce the second messenger compound cAMP [17,18,19] to dissociate Bcy1 from an inhibitory complex with either Tpk1, Tpk2, or Tpk3 (the three PKA catalytic proteins), thus activating PKA [20,21]. One of the major downstream roles of PKA signaling (via Yak1) is to inhibit the nuclear localization of the stress response transcription factors Hsf1, Msn2, and Msn4 [22].

Another second messenger in response to glucose is increased cytoplasmic alkalinity [23,24]. The yeast plasma membrane ATPase (Pma1) is the most abundant plasma membrane protein and the largest consumer of ATP [25]. The plasma membrane (Pma1) and vacuolar (Vma1) ATPases maintain a slightly alkaline cytoplasmic pH of ~7.8 by pumping protons out of the cell and into the vacuole, respectively [26]. When cells are starved for carbon, these pumps are inactivated by dephosphorylation by Glc7 [27] and by inhibition by Hsp30 [28]. This results in a rapid acidification of the intracellular space to pH ~5.7 [29,30]. The decrease in intracellular pH is crucial for viability upon carbon starvation and is thought to conserve energy, leading to storage of metabolic enzymes (such as Gln1) in filamentous assemblies [31], and to decreased membrane biogenesis [32]. Cytoplasmic acidification also inhibits Cyr1 (adenyl cyclase) and thus PKA activity [24] via inactivation of the vacuolar ATPases [24,29].

Trafficking of Pma1 to the plasma membrane involves Exp1 and Psg1. In the absence of Exp1, Pma1 is retained in the endoplasmic reticulum, whereas deletion of *PSG1* increases Pma1 degradation in the vacuole [33]. In the presence of glucose, Pma1 activity increases ten-fold by phosphorylation of the C-terminal tail (at S899) by Ptk2 (with a smaller contribution from Hrk1). This increases the affinity of Pma1 for ATP, and thus its activity. In addition, the S911/T912 amino acids of Pma1 are also phosphorylated by an unknown kinase, increasing its Vmax [34,35]. Indeed, truncation of the final 18 amino acids of Pma1 (Pma1-Δ901) increases Pma1 activity of cells lacking glucose as if glucose were present [36]. However, how Ptk2 and Hrk1 are regulated by glucose remains undetermined.

## 3. Snf1 Structure

The SNF1/SnRK/AMPK complex is (generally) composed of three proteins. Snf1/AMPKa (the α-subunit) harbors the complex’s kinase activity and is itself composed of a kinase domain (amino-acids 1-391, KD) and a regulatory domain (amino-acids 392-633, RD) [37] (Figure 1). In the absence of glucose, one of three β-subunits (Sip1, Sip2, Gal83) binds to aa515-633 (RD-β) of Snf1 [38]—these regulate the localization of the SNF1 complex. In the presence of glucose, these proteins are cytoplasmic but in the absence of glucose, Sip1 is enriched at the vacuolar membrane, Sip2 remains cytoplasmic (and is myristoylated at the plasma membrane), and Gal83 enters the nucleus [39]. Gal83 is required for nuclear activity of SNF1 [39]. Overexpression of Sip1 improves cell survival upon exposure to 6% ethanol by 21 ± 2% [40] The β-subunits of mammalian AMPK also interact with AMPK/Snf1 at the far-N terminal region to form the ADaM pocket [41,42]—we have termed this region the pre-kinase region (PKR, aa1-53) (Figure 1) and in *S. cerevisiae*, certainly the distal portion of this region is required for kinase activity [43]. The PKR is diverse in amino acid sequence, whereas the succeeding kinase domain is remarkably conserved across kingdoms. In fungi, the PKR is host to monoamino-acid repeats, with *Saccharomyces* and *Candida* species often possessing polyhistidine tracts. The regulatory domain contains an auto-inhibition domain (AID) (aa392-495—RDγ, Figure 1) that interacts with, and inhibits, kinase activity [37], with subsequent work narrowing the AID to being aa 460-495 [44]. Snf4/AMPKc (the γ-subunit) is a regulatory protein that, in the absence of glucose, binds to the AID preventing the auto-inhibition [37]; if the AID is deleted, then Snf4 is dispensable for Snf1 activity in the absence of glucose [45]. Snf1 must associate with Snf4 and one of the beta proteins for a stable, active complex to form [46]. Interestingly, in some filamentous fungi (e.g., *Cochliobolus carbonum*), the β-subunit has fused with Snf1; other fungi have differing numbers of β-subunit proteins. A schematic diagram of Snf1 structure is presented as Figure 1.

In addition to glucose deprivation, Snf1 is also activated by other stresses, including arsenite [47], selenite (glutathione oxidation), [48] sodium and lithium ion stress [49], high osmolarity [50], hydroxyurea [51], media alkalinity [52], and oxidative stress (peroxide) [50]. However, how Snf1 integrates the signals from these stresses has not yet been elucidated.

## 4. Regulation of Snf1

### 4.1. Phosphorylation

Since Snf1 activation leads to a global rewiring of metabolism and transcription, it would be expected to be under tight regulation by glucose. Most protein kinases become activated by being phosphorylated by an upstream protein kinase, and Snf1 is no exception. The classical mechanism by which *S. cerevisiae* Snf1 is regulated is its activation by phosphorylation by three kinases (Sak1, Elm1, Tos3) at threonine 210 in the activation loop [53,54], of which Sak1 contributes the greatest share of the phosphorylation [50], although Tos3 has been reported to be important for maintenance of Snf1 catalytic activity and for the expression of gluconeogenesis genes during prolonged growth on glycerol/ethanol (as opposed to acute carbon source switching) [55]. In the presence of glucose, PKA can somewhat inhibit Sak1 [56], although the inhibition of Snf1 by glucose does not require the PKA pathway [57]. Phosphorylation at T210 of *S. cerevisiae* Snf1 was discovered long ago, in 1992 [58], and has been extensively researched. In addition, and as expected from a protein kinase activated by protein kinases, a second level of regulation is carried out by phosphatases that take out the activating signal. Snf1 kinase is dephosphorylated and inactivated by type 1 protein phosphatase Glc7-Reg1 when glucose is present at high concentration [59]—other phosphatases can also contribute to Snf1 dephosphorylation such as Glc7-Reg2 [60] after prolonged glucose starvation [61], Sit4 [62], Ptc1 [63], and Ppz1 (when overexpressed) [64]. Nevertheless, Glc7-Reg1 is the predominant phosphatase and in a *reg1Δ* cell, Snf1 is phosphorylated and active in glucose (59). ADP binding to Snf4 protects Snf1-T210 from dephosphorylation by Glc7 (Figure 2A); ADP binding buries the activation loop inside the SNF1 complex by fostering an interaction with a conserved histidine in the β-subunits (Gal83^H379^, Sip2^H375^) [65]. This is in contrast to metazoan AMPK where AMP protects against dephosphorylation of T172 (corresponding to yeast’s T210) [66]. However, an alternative model posits that it is the interaction of the kinase domain with the interface between the three proteins that comprise an active SNF1 complex that activates Snf1 kinase activity, with AMP binding at the kinase active site protecting phosphorylated T210 from attack by its phosphatases [67], as evidenced by the lack of dephosphorylation when 2NM-PP1 binds to the analogue sensitive Snf1^I132G^ mutant [16,67]. Thus, we can see that three-dimensional changes in the protein architecture play a role in ensuring the activity of the complex, once the kinase is activated.

### 4.2. SUMOylation

SUMO (Smt3, Small Ubiquitin-like MOdifier) is a ubiquitin-like 101 amino acid protein that is conjugated to target lysines by one of three SUMO E3-ligases (Siz1, Siz2, and Mms21). As its name implies, SUMO is a ubiquitin-like protein that carries out a plethora of signaling jobs within the cell. As well as modifying a target of the protein structure, its main property is to act as a “molecular glue” via interaction with SUMO Interacting Motifs (SIMs) [68]. This allows to increase the concentration of particular proteins at a particular location, promoting the activity of some and/or facilitating protein–protein interactions. SUMOylated proteins can also be recognized by the Slx5-Slx8 ubiquitin ligase (a StUbL—SUMO-targeted Ubiquitin Ligase) which forwards it for proteasomal degradation [69]. Snf1 activity is inhibited by SUMOylation of Snf1 at a non-canonical SUMOylation site at lysine 549 (K549) by the Mms21 SUMO E3-ligase [16,70] (Figure 2B). Mms21 is activated by phosphorylation at S261 by PKA in response to glucose, and by Mec1 in response to DNA damage [70]. SUMOylation of Snf1 has two effects: inhibition of Snf1 by interaction with a SUMO Interacting Motif (SIM) at I219, thus leading to contact between the far-C terminal region and the kinase domain; and also ubiquitination of Snf1 by Slx5-Slx8 leading to degradation of Snf1 [16]. The ubiquitination of SUMOylated Snf1 is reversed by Ubp8 [16,71] and Snf1 is deSUMOylated by Ulp1 [16]. Thus, SUMO plays a central role in maintaining appropriate levels of the protein, as well as changing the 3D structure of the protein to prevent further activity. SUMOylation of Snf1 has since also been observed in *A. thalania* (SnRK1) [72] and mice (AMPKα1) [73], suggesting that this type of regulation may be widely conserved.

### 4.3. Regulation of Snf1 by pH

Histidine is both an aromatic and a positively charged amino acid. Deprotonated histidine can engage in π−π and hydrogen–π interactions with other aromatic amino acids [74]. The pK_a_ of the histidine sidechain is 6.9 [75], which could enable this residue to function as an intracellular pH sensor. Celenza et al. (1989) noted the presence of a sequence comprising 13 consecutive histidine residues followed by a glycine and an additional histidine at amino acids 18–32 in the pre-kinase region (PKR) at the far N-terminal of Snf1 [76] (Figure 1), and reported that deletion of 12 of these histidines did not inhibit secreted invertase (Suc2) activity [76]. Indeed, deletion of this polyhistidine tract has the opposite effect: it increases Snf1 activity by 50% (as determined by *ADH2, ACS1*, *JEN1* [43,77], and *SUC2* [77] expression). Replacement of these 14 histidine residues with alanine results in the same increase in Snf1 activity, whereas replacement with the aromatic amino acids phenylalanine, tyrosine, or tryptophan ablates Snf1 activity. Deletion of *HSP30* [28] and hyper-activation of Pma1 by truncation of its C-terminal tail [36] mimic high glucose conditions and are required for the maintenance of lower Snf1 activity even upon glucose deprivation [43], whereas deletion of *EXP1* (thus lowering Pma1 presence at the plasma membrane [33]) prolongs Snf1 activity upon glucose replenishment [43]. Thus, in the presence of glucose, the deprotonated polyhistidine tract interacts with the β-subunit binding site, precluding interaction with Sip1, Sip2, or Gal83 and thus inhibiting Snf1 activation and translocation to the nucleus [43]. Under these circumstances, again the 3D conformation of the kinase is in a “closed” position. In contrast, in the absence of glucose, the lowering of Pma1 proton export activity results in a decrease in intracellular pH [23,34], which causes protonation of the polyhistidine tract and its dissociation from the β-subunit binding site and thus permits SNF1 complex formation [43] (Figure 2C). The polyhistidine tract is progressive, with decreased inhibition of Snf1 with fewer histidine residues [43]. We interpret this result as implying that the sensing of pH by Snf1 arises from the aggregate changes in the charge of the polyhistidine tract. This in turn implies that the mechanism by which pH may be sensed requires π-bond stacking for the interaction with the β-subunit binding site, at Snf1 regulatory domain.

## 5. The Polyhistidine Tract Is a Multi-Tool

The dual aromatic/charged nature of histidine is exemplified by the discovery of a further interaction that requires protonated histidine. The interaction between polyhistidine and the β-subunit binding site of Snf1 occurs in the presence of glucose when the polyhistidine stretch is deprotonated [43]. In contrast, in the absence of both glucose and iron, the polyhistidine stretch becomes protonated and interacts with an acidic/negatively polar patch of the iron regulator Aft1 (aa16-24). This interaction inhibits nuclear Snf1 activity by 50% [77]. Both the ironscarcity-induced transcription factor Aft1 (but not its paralogue Aft2) and Snf1 need to be localized to the nucleus for Aft1 to inhibit Snf1, with the interaction occurring at the nuclear membrane [77]. Thus, when iron becomes scarce and several cellular processes that require this metal for normal function (such as DNA metabolism) compete, the activity of Snf1 is inhibited. However, only nuclear activity becomes low, and cytosolic activities continue as usual. The spatial and temporal regulation of Snf1 activity enables not only to respond to carbon source availability, but also to exert a differential response to iron depending on the type of carbon source available. Moreover, since iron–sulfur clusters are produced in the mitochondria, coordinating nuclear Snf1 activity to iron availability ensures that sufficient clusters are available to support the enzymatic activities needed for respiration, while testing mitochondrial competency before nuclear Snf1 is activated. This type of interaction to further modify Snf1 activity due to an additional stress condition may also account for how Snf1 integrates signals from multiple stresses, including others for which no mechanism is known.

## 6. Other Functions of Polyhistidine

Although two functions of the polyhistidine tract of Snf1 have been identified, this does not preclude additional potential roles. Consecutive histidine residues are very rare. In *S*. *cerevisiae*, only 12 other proteins contain polyHIS stretches with more than 7 histidines (H^7+^), representing 0.25% of the *S*. *cerevisiae* proteome (assuming 5815 proteins [78]), and the human genome contains 86 polyH^7+^ proteins [79], representing 0.42% of the proteome (assuming 20,352 protein-encoding genes [80]). The proteins containing polyH^7+^ in yeast are diverse and do not seem to have a common function, although 4 of the 12 are involved in metabolism and stress responses (*Snf1*, *Tax4*, *Rom2*, *Bag7*). However, in humans, there is a disproportionate number of proteins with H^7+^ that function as voltage-gated channels—this could permit a progressive change in voltage to accumulate in the histidines and when a threshold is reached to effect a conformational/interactional change. Indeed, multiple histidine residues have been found to be important in the snail *Helisoma trivolvis* voltage-gated channel Hv1 [81,82] but these have not been investigated together. The protonation of four out of the seven histidines in Snf5 (these are dispersed throughout a glutamine rich low-complexity region) is involved in the activation of the *SWI/SNF* complex to express *ADH2* in the absence of glucose [83]. The deprotonation of a single histidine residue (H212) in vertebrate RasGRP1 suffices to increase basal activity, whereas a constitutive positive charge (H212K) lowers activity [84]. The use of histidine residues for sensing pH is thus widespread, though the case of *S. cerevisiae* Snf1 is the first reported instance of polyhistidine as a pH meter (in the sense that it responds progressively to both increased and decreased levels).

Polyhistidine tracts of proteins of other organisms mediate protein–protein interactions. In humans, for example, the mitogenic growth factor Granulin binds to the polyhistidine tract of cyclin T1. This inhibits cyclin T1-Cdk9 activity towards the C-terminal phosphorylation sites of RNA polymerase II [85]. Many proteins undergo phase separation; the polyhistidine tract of Yin Yang protein 1 (YY1) mediates its phase separation and association with many other proteins such as the transcription factors EP300, BRD4, MED1, FOXM1, and active RNA polymerase II, and is required for the oncogenicity of this protein [86]. Indeed, of the proteins in the human proteome that possess a polyhistidine tract greater than 5 histidines; 72 of them are targeted to nuclear speckles, with an enrichment within this population for transcription factors and RNA processing proteins [79].

Polyhistidine is also involved in metal binding in vivo. The polyhistidine tracts of the nuclear speckle [79] localized proteins FOXG1B and MAFA (human) binding to Cu^2+^ and Zn^2+^ [87], though it is not determined if this metal binding competes with the nuclear speckle localization or not. The peptic ulcer-causing bacterium *Helicobacter pylori* highly expresses (2% of total protein) the 60 amino acid Hpn protein, which contains 28 histidine residues as polyhistidine tracts [88]. The internal pH of *H. pylori* is 7.5–7.75 [89], so these histidines would be expected to be predominantly deprotonated. Histidines in this protein bind to Cu^2+^ and Ni^2+^ [88], so such a metal interaction cannot be precluded from occurring with Snf1.

Polyphosphate is a chain of three to thousands of phosphates, and occurs in all kingdoms of life [90]. Neville et al. has shown that 19 of 27 selected human polyhistidine-containing proteins ionically bind polyphosphate, and 10 *S. cerevisiae* proteins containing polyhistidine also bind polyphosphate [91] (including Snf1), and show a dramatic shift on neutral pH (NuPAGE) acrylamide gels. For Snf1^1−65^, this interaction is specific for histidine since substitution to arginine or alanine abolishes the shift [91]. However, it remains to be determined if the interactions seen in the gel are of physiological relevance to Snf1 in vivo since polyphosphates are supposed to be restricted to the vacuole.

## 7. Relationship between the Snf1-Activating Pathways

In the presence of glucose, the auto-inhibition domain inhibits Snf1 kinase activity by interacting with the kinase domain—this is relieved by phosphorylation at T210 and by Snf4 binding to the auto-inhibition domain [45]. SUMOylated K549 interacts with the kinase domain at the SIM centered on I129 [16]. Finally, the polyhistidine tract interacts with the β-subunit binding site preventing Sip1/Sip2/Gal83 binding [43]. Thus, each means of regulating Snf1 acts upon a different part of the Snf1 protein; this is illustrated in Figure 2.

The three activation mechanisms are also distinct. Phosphorylation of T210 is regulated by ADP, preventing access of phosphatases to phosphorylated T210 [65]; SUMOylation acts by PKA activation of Mms21 [70]; and protonation of the polyhistidine tract is regulated by Pma1 activity [43] (which is upregulated by Ptk2 and Hrk1 kinases in the presence of glucose [34,35]). However, above Snf1, there is extensive cross talk between the glucose-sensing pathways. PKA (activated by glucose) and Snf1 (activated by glucose deprivation) are mutually antagonistic, with Snf1 inhibiting PKA by phosphorylating Cyr1 (adenyl cyclase) at four locations [8]. The negative regulation of Snf1 by PKA occurs by three mechanisms: 1. By activating Mms21 to SUMOylate and inhibit Snf1 [70]. As we have seen above, SUMOylation of Snf1 at lysine 549 inhibits Snf1 by changing its shape and sending it to degradation (after ubiquitination). 2. By lowering Sak1 activity towards Snf1 T210 [56]. As T210 phosphorylation is the main activating mechanism for Snf1, the reduction of activity of Sak1 shifts the cells to fermentative metabolism. 3. PKA phosphorylates Sip1 [56] to prevent localization of the SNF1 complex to the vacuole [92]. However, it should be emphasized that PKA activity is not required for the glucose inhibition of Snf1 [57]. Furthermore, the antagonism between Snf1 and PKA extends to glucose sensing by Snf3/Rgt2; with non-dephosphorylated T172 [15,16] or unSUMOylated Snf1 [16] inhibiting Mth1 degradation even in the presence of glucose whereas inactivation of the Rgt1 repressor requires PKA activity [13]. Similarly, cytoplasmic acidification activates Snf1 [43] whilst simultaneously inhibiting PKA by inactivating vacuolar ATPases [24,29]. Thus, despite the lower research attention paid to pH regulation of the glucose response, cytoplasmic acidification does appear to be the key master regulator of the different signaling responses to glucose.

At the Snf1 molecule level, the three regulatory systems are independent from each other. Prevention of SUMOylation of Snf1 (Snf1^K549R^ or Mms21-CH) or the interaction of SUMOylated Snf1 with the I129 SIM (Snf1^I129A^) does not prevent dephosphorylation of Snf1 T210 in the presence of glucose [70]. Mutation of the polyhistidine tract to aromatic amino acids (abolishing Snf1 activity), deletion or alanine substitution (hyperactivating Snf1), or increasing Pma1 activity (deprotonating the polyhistidine tract) do not affect phosphorylation at T210—this also demonstrates that phosphorylation at T210 does not always correlate with Snf1 activity, and thus should not be used as the sole readout of Snf1 [43]. Furthermore, deletion of the polyhistidine tract does not compensate for a T210A mutation, showing that both of these regulatory mechanisms are independent [43]. Similarly, the strong interaction of the kinase and regulatory domains of Snf1 when the polyhistidine tract is substituted with aromatic amino acids does not compensate for lack of SUMOylation at K549 [43].

Thus, despite the extensive cross-talk between the glucose-sensing pathways, at the level of the Snf1 protein itself each of the three regulatory mechanisms discussed are independent and can each inhibit Snf1. This separation is likely to provide for optimal fermentation of any glucose/fructose present in the environment. In addition, the multiplicity of independent regulations allows Snf1 to respond to a variety of stresses and metabolic cues, in addition to carbon sources.

## 8. Future Directions

We have shown that the Snf1/AMPK protein kinase plays, together with PKA, a central role in the decision between fermentation and respiration, depending on the availability of glucose or other, less preferred, carbon sources. However, in the last years, it has become apparent that Snf1 can respond to additional environmental signals and participate in the still unclear mechanisms of catabolite repression. Future work should investigate how Snf1 can respond to additional environmental cues and how it helps coordinating them with carbon source availability. We have shown one such example with the coordination between sugars and iron, monitored and coordinated by the polyhistidine tract. Further research is needed to unveil other levels of regulation of this fascinating central metabolic kinase.

## Figures and Tables

**Figure 1 biology-12-01007-f001:**
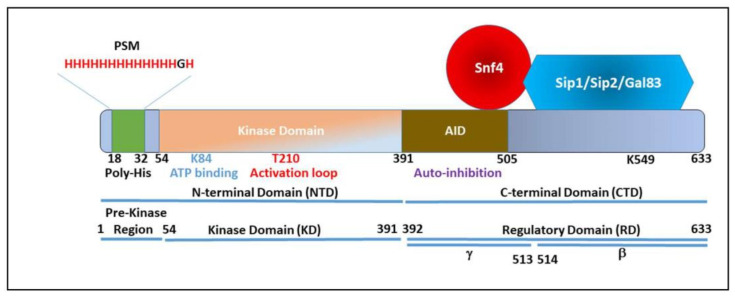
A schematic diagram showing the domains of Snf1 and motifs/amino acids that are discussed in this review.

**Figure 2 biology-12-01007-f002:**
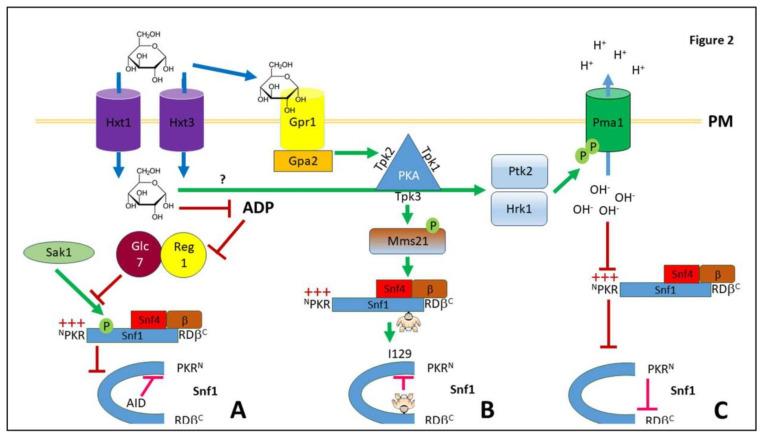
An illustrative diagram to show the three independent mechanisms by which glucose inhibits Snf1 activity: (**A**). Regulation by dephosphorylation of T210. Glucose enters the cell through glucose transporters. Glucose metabolism decreases the amount of ADP in the cell, thus permitting dephosphorylation of T210 by Glc-Reg1 and other phosphatases. This leads to dissociation of Snf4 and the β-subunit of the SNF1 complex, and auto-inhibition of the Snf1 kinase domain. Elm1 and Tos3 are omitted for clarity. (**B**). Regulation by SUMOylation. Glucose is sensed by the Gpr1 receptor which activates the Gpa2 G-protein, leading to activation of PKA (intermediate steps omitted for clarity). PKA activates the SUMO E3-ligase Mms21 by phosphorylating S261 (this site is also phosphorylated by Mec1/Tel1 following DNA damage). Mms21 SUMOylates Snf1 at K549; the SUMO moiety interacts with a SIM at I129 to inhibit Snf1. (**C**). Regulation by pH. Glucose activates the kinases Ptk2 and Hrk1 by an unknown mechanism, which upregulates Pma1 protein-export by phosphorylating Pma1 at S899. Increased cytoplasmic alkalization deprotonates the polyhistidine tract within the PKR resulting in interaction of the polyhistidine with RD-β preventing β-subunit binding to Snf1.

## Data Availability

Not applicable.

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
