# Peer review of "Glucose Inhibits Yeast AMPK (Snf1) by Three Independent Mechanisms"

_biology, 2023, doi:10.3390/biology12071007_

Round 1

Reviewer 1 Report

A well written comprehensive review presenting the versatile role of Snf1/AMPK protein kinase in relation to respiratory of fermentative glucose metabolism. Moreover, it might be valuable to include potential effects of ethanol on Snf1/AMPK protein kinase in the mansucript.

Reviewer 2 Report

The authors Simpson and Kupie present a well rounded manuscript that review the glucose sensing mechanisms, shedding light on Snf1 inhibition. 
Apart from the the introduction (lines 24-42) the text is well described and is easily understandable. Nonetheless, please check the text again for recurring double blanks and missing commas.
The introduction itself (lines 24-42) needs to be split up into more sentences. The writers use too many semicolons and commas without a full stop of the sentence leading to unreadability of this part.

Issues with  the text:
Missing commas, too long sentences, and double blank errors. 

Reviewer 3 Report

This review by Kobi Simpson and Martin Kupiec is a very good recapitulation of the different mechanisms of regulation of a very relevant kinase such as Snf1/AMPK. The two authors are outstanding researchers in the filed and have described themselves some of the mechanisms reviewed in this paper.

However, the organization of the review could be improved, rearranging some paragraphs:

1)    Lines 54-59: the authors describe the Cyr1/PKA pathway; I would suggest to:

·       move here all the part about the crosstalk between Snf1 and PKA (see also point 5),

·       mention here that Snf1 phosphorylates Cyr1 regulating this pathway,

·       add the data showing that glucose-dependent inactivation of Snf1/AMPK is independent from the Ras/PKA pathway (see DOI: 10.3390/ijms22179483).

2) Lines 132-135: the authors mention the role of ADP in protecting Snf1-T210 from dephosphorylation, citing the paper by Mayer et al., 2011. I would suggest including also the data by Chandrashekarappa et al, 2013 (doi: 10.1074/jbc.M112.422659), proposing an alternative explanation of the role of ADP in Snf1 regulation.

3) Paragraph 5 should be included in paragraph 4, point 3).

4) Paragraph 6, although interesting, is not directly linked to Snf1 regulation.  

5) Paragraph 7: Data about PKA are already present in the previous paragraphs, I would remove this paragraph and expand the different concepts in the proper paragraphs (i.e. move PKA regulating SUMOylation in paragraph 4 part 2; move PKA regulating phosphorylation in paragraph 4 part 1; move regulation of Snf1 on Cyr1 in paragraph 2). This change would give greater emphasis on the regulation described above.

Generally the English language is good. Some typos are present, please correct them (missing punctuation, double spaces).
